# Is Part-of-Speech Tagging a Solved Problem for Icelandic?

**Örvar Kárason**
Department of Computer Science
Reykjavik University
Iceland
orvark13@ru.is

**Hrafn Loftsson**
Department of Computer Science
Reykjavik University
Iceland
hrafn@ru.is

## Abstract

We train and evaluate four Part-of-Speech tagging models for Icelandic. Three are older models that obtained the highest accuracy for Icelandic when they were introduced. The fourth model is of a type that currently reaches state-of-the-art accuracy. We use the most recent version of the MIM-GOLD training/testing corpus, its newest tagset, and augmentation data to obtain results that are comparable between the various models. We examine the accuracy improvements with each model and analyse the errors produced by our transformer model, which is based on a previously published ConvBERT model. For the set of errors that all the models make, and for which they predict the same tag, we extract a random subset for manual inspection. Extrapolating from this subset, we obtain a lower bound estimate on annotation errors in the corpus as well as on some unsolvable tagging errors. We argue that further tagging accuracy gains for Icelandic can still be obtained by fixing the errors in MIM-GOLD and, furthermore, that it should still be possible to squeeze out some small gains from our transformer model.

## 1 Introduction

Part-of-Speech (POS) tagging is a sequential labelling task in which each token, i.e., words, symbols, and punctuation in running text is assigned a morphosyntactic tag. It is an important step for many Natural Language Processing applications. A token is ambiguous when it has more than one possible tag. The source of ambiguity is polysemy in the form of homographs from the same word class, from different word classes, and also within the declension paradigms of the same word. The task, therefore, entails examining the token itself and its context for clues for predicting the correct tag. For the last mentioned type of ambiguity, which is prevalent in Icelandic, it is necessary to find another unambiguous token in the context that the target token shows agreement with and use it to determine the correct target tag.

Over the last two decades, steady progress has been made in POS tagging for Icelandic. Various taggers have been presented throughout this period that improved on previous state-of-the-art (SOTA) methods (Rögnvaldsson et al., 2002; Helgadóttir, 2005; Loftsson, 2008; Dredze and Wallenberg, 2008; Loftsson et al., 2009, 2011; Loftsson and Östling, 2013; Steingrímsson et al., 2019; Snæbjarnarson et al., 2022; Daðason and Loftsson, 2022; Jónsson and Loftsson, 2022).

Work on Icelandic corpora has also progressed. Existing corpora have undergone error correction phases (Barkarson et al., 2021), and, in some cases, been expanded with new data (Barkarson et al., 2022). A new larger gold standard corpus for POS tagging, *MIM-GOLD* (Loftsson et al., 2010), was created to replace the older standard, the *Icelandic Frequency Dictionary* (IFD, Pind et al. 1991), and multiple alterations have been made to the fine-grained Icelandic tagset (Steingrímsson et al., 2018; Barkarson et al., 2021).

All this variability over the years means that previously reported results for POS taggers are not easily comparable. Thus, we train and test four data-driven taggers that have been employed for Icelandic (see Section 3), using the latest version of MIM-GOLD and its underlying tagset, as well as the latest versions of augmentation data (see Section 2). We obtain SOTA tagging accuracy by training and fine-tuning a ConvBERT-base model in a slightly different manner than previously reported by Daðason and Loftsson (2022) (see Section 3).

With the latest tagging method based on the transformer model finally reaching above 97% per-token accuracy for Icelandic (Jónsson and Loftsson, 2022; Snæbjarnarson et al., 2022; Daðason and Loftsson, 2022), the generally believed limit of inter-annotator agreement (Manning, 2011), we might ask ourselves if POS tagging is now a solved problem for Icelandic. Indeed, our evaluation results show that the tagging accuracy of our ConvBERT-base model is close to 98% (see Table 3). A large portion of the remaining errors can be explained by 1) a lack of context information to make the correct prediction, and 2) annotation errors or other faults in the training/testing corpus itself. Addressing the latter should give further gains. Furthermore, some small additional gains could be squeezed out of the transformer model, by using a larger model and pre-training it on more data. When this is done, we may be able to argue that POS tagging is a solved problem for Icelandic.

The rest of this paper is structured as follows. In Sections 2 and 3, we describe the data and the models, respectively, used in our experiments. We present the evaluation results in Section 4, and detailed error analysis in Section 5. Finally, we conclude in Section 6.

## 2 Data

In this section, we describe the data and the tagset used in our work.

### 2.1 Corpus

The MIM-GOLD corpus is a curated subset of the MIM corpus (Helgadóttir et al., 2012) and was semi-automatically tagged using a combination of taggers (Loftsson et al., 2010). Version 21.05 of the corpus contains 1 million running words from 13 different text types, of which about half originate from newspapers and books (see Table 1). All versions of MIM-GOLD include the same 10-fold splits for use in cross-validation.[1]

MIM-GOLD was created to replace the IFD as the gold standard for POS tagging of Icelandic texts. The IFD corpus was sourced from books published in the eighties and has a clear literary and standardized language slant. Steingrímsson et al. (2019) reported a 1.11 percentage point (pp)

| Text type | % of all |
|---|---|
| Newspaper *Morgunblaðið* | 24.9 |
| Books | 23.5 |
| Blogs | 13.4 |
| Newspaper *Fréttablaðið* | 9.4 |
| The Icelandic Web of Science | 9.1 |
| Websites | 6.5 |
| Laws | 4.1 |
| School essays | 3.4 |
| Written-to-be-spoken | 1.9 |
| Adjudications | 1.3 |
| Radio news scripts | 1.1 |
| Web media | 0.8 |
| E-mails | 0.5 |
| Total | 100.0 |

Table 1: Information about the various text types in MIM-GOLD, adapted from Loftsson et al. (2010).

lower per-token accuracy for MIM-GOLD compared to the IFD.

### 2.2 Morphological lexicon

Version 22.09 of the Database of Modern Icelandic Inflection (DMII) (Bjarnadóttir, 2012), which is now a part of the Database of Icelandic Morphology (Bjarnadóttir et al., 2019), contains 6.9 million inflectional forms and about 330 thousand declension paradigms.[2] Though the database cannot be used directly to train a POS tagger, as there is no context or distributional information for the word forms, it has been used to augment taggers during training and help with tagging unknown words (words not seen during training) (Loftsson et al., 2011; Steingrímsson et al., 2019).

### 2.3 Pre-training corpus

The Icelandic Gigaword Corpus (IGC), which includes text sources from multiple varied domains, has been expanded annually since 2018 (Barkarson et al., 2022). The motivation for constructing the IGC was, *inter alia*, to make the development of large Icelandic language models possible (Steingrímsson et al., 2018). The 2021 version used in our work contains about 1.8 billion tokens.[3]

---

[1]Version 21.05 is available at `http://hdl.handle.net/20.500.12537/114`

[2]`https://bin.arnastofnun.is/DMII/LTdata/`

[3]Version 2021 is available at `http://hdl.handle.net/20.500.12537/192`

## 2.4 Tagset

The MIM-GOLD tagset v. 2 is the fourth iteration of the fine-grained tagset that is exclusively used for modern Icelandic and has its origin in the IFD. The tagset consists of 571 possible tags, of which 557 occur in MIM-GOLD.

The tags are morphosyntactic encodings consisting of one to six characters, each denoting some feature. The first character denotes the *lexical category* and is, in some cases, followed by a sub-category character. For each category, a fixed number of additional feature characters follow, e.g., *gender*, *number* and *case* for nouns; *degree* and *declension* for adjectives; and *voice*, *mood* and *tense* for verbs. To illustrate, consider the word form *konan* ('the woman'). The corresponding tag is *nveng*, denoting noun (*n*), feminine (*v*), singular (*e*), nominative (*n*) case, and definite suffixed article (*g*).

## 3 Models

In this section, we describe the four data-driven POS tagging models we trained and evaluated:

- **TriTagger** (Loftsson et al., 2009) is a reimplementation of TnT (Brants, 2000), a second order (trigram) Hidden Markov model. The probabilities of the model are estimated from a training corpus using maximum likelihood estimation. Assignments of POS tags to tokens is found by optimising the product of lexical probabilities ($p(w_i|t_j)$) and contextual probabilities ($p(t_i|t_{i-1}, t_{i-2})$) (where $w_i$ and $t_i$ are the $i^{th}$ word and tag, respectively).

  When work on creating a tagger for Icelandic started at the turn of the century, five existing data-driven taggers were tested on the IFD corpus (Helgadóttir, 2005). TnT obtained the highest accuracy and has often been included for comparison in subsequent work.

- **IceStagger** (Loftsson and Östling, 2013) is an averaged perceptron model (Collins, 2002), an early and simple version of a neural network.[4] It learns binary feature functions from predefined templates. The templates are hand-crafted and can reference adjacent words, previous tags, and various custom matching functions applied to them. The

templates, intended to capture dependencies specific to Icelandic, were developed against the IFD. During training, the algorithm learns which feature functions are good indicators of the assigned tag, given the context available to the templates. It does that by adjusting the weight associated with the feature function. The highest-scoring tag sequence is approximated using beam search. Both IceStagger and TriTagger use data from the DMII to help with guessing the tags for unknown tokens.

- **ABLTagger v. 1** (Steingrímsson et al., 2019; Jónsson and Loftsson, 2022) is based on a bidirectional long short-term memory (Bi-LSTM) model.[5] That model is an extension of LSTMs (Hochreiter and Schmidhuber, 1997) that can be employed when the input is the whole sequence. Two LSTMs are trained on the input, with the second traversing it in reverse (Graves and Schmidhuber, 2005). The input for ABLTagger consists of both word and character embeddings. The model is augmented with n-hot vectors created from all the potential lexical features of the word forms from the DMII. ABLTagger was developed against the IFD but was the first tagger to be applied to MIM-GOLD.

- **ConvBERT** (Jiang et al., 2020) is an improved version of the BERT model (Vaswani et al., 2017; Devlin et al., 2019) that is more efficient and accurate. We used an existing ConvBERT-base model pre-trained on the IGC by Daðason and Loftsson (2022)[6] and fine-tuned it for tagging on MIM-GOLD. This is a standard pre-trained transformer model with two changes: the embeddings of the first and last subwords are concatenated (`first+last` subword pooling) to generate the token representations (Schuster and Nakajima, 2012), and we continued the pre-training of the ConvBERT-base model using the training data of each fold from MIM-GOLD for three epochs before fine-tuning it for tagging for 10 epochs with the same data. Each modification gave a 0.07 pp

---

[4]IceStagger and TriTagger are included in the IceNLP toolkit (Loftsson and Rögnvaldsson, 2007): `https://github.com/hrafnl/icenlp`

[5]ABLTagger v. 1 is available at `https://hdl.handle.net/20.500.12537/53`

[6]`https://huggingface.co/jonfd/convbert-base-igc-is`

| | Token acc. | Sent. acc. |
|---|---|---|
| TriTagger | 91.01% | 35.58% |
| IceStagger | 92.72% | 42.74% |
| ABLTagger v1 | 94.56% | 49.11% |
| **ConvBERT-base** | **97.79%** | **73.43%** |

Table 2: Token and sentence tagging accuracy for the four models.

improvement in accuracy; i.e. 0.14 pp in to-tal.[7]

# 4 Results

We evaluated the four models by applying 10-fold cross-validation (CV) using the standard splits in MIM-GOLD (see Section 2). The results are shown in Table 2. The transformer model, ConvBERT-base, obtains 6.78 pp higher accuracy than the HMM model (TriTagger), which is equivalent to a 75.42% reduction in errors!

The increase in sentence accuracy, which is often overlooked, is also very impressive. It has more than doubled and now close to $\frac{3}{4}$ of the sentences are correct. Sentences come in different lengths, ranging from a single token up to 1,334 tokens in MIM-GOLD, and increased length can result in increased complexity. Figure 1 shows the length distribution of sentences with no errors. The figure shows both general accuracy gains as well as an improvement in handling longer sentences.

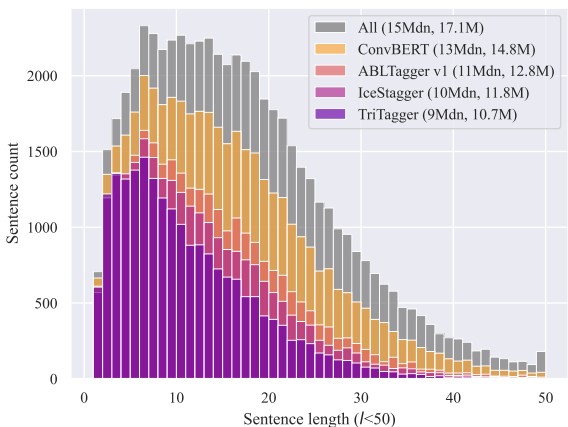

Figure 1: Distributions of correctly tagged sentences. The legend shows each set's median (Mdn) and mean (M).

[7]See https://github.com/orvark13/postr/ for training and evaluation scripts, as well as fine-tuned models.

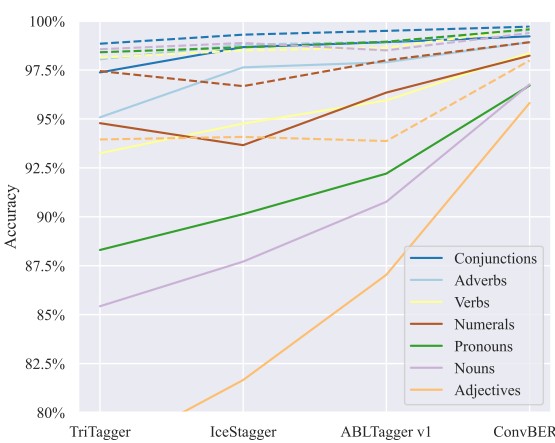

Figure 2: The accuracy improvements between the models for the more frequent lexical categories. Solid lines are the per-token accuracy for all tags in that category, and dashed lines are the lexical class accuracy, i.e., the tag category is correct but there is some error in the predicted features. Errors within the categories diminish as those lines converge.

## 4.1 Accuracy improvements

TriTagger and IceStagger are limited to a three-token window and they need frequency information of tokens to learn from. As is to be expected, IceStagger gains accuracy according to the feature templates pre-defined for it. ABLTagger's improvements come from the BiLSTM's context window being the whole sentence and it, thereby, being able to detect long-range dependencies. Its ability to see within the token by means of the character embeddings helps it handle tokens not seen during training. Augmenting the model with data from DMII also helps with unknown words.

The source of improvement for the transformer model is mainly threefold. First, the attention mechanism aids it in selecting the right dependencies (e.g., when there is more than one option), and it is detecting longer long-range dependencies than the BiLSTM model. We see this from the examination of the predictions and it is also indicated by the model's success with longer sentences as is evident in the shape of its distribution in Figure 1. Secondly, the model is often able to discern the different semantic senses of ambiguous tokens. We assume this stems from the contextual word embeddings in the large pre-trained Conv-BERT language model. Finally, it benefits from all the language sense from the IGC infused in the

| POS Transformer Model | Accuracy |
|---|---|
| IceBERT-IGC [1] | 97.37% |
| ConvBERT-base [1] | 97.75% |
| *Our ConvBERT-base* | **97.79%** |
| Excluding *x* and *e* tags | |
| IceBERT-IGC, multi-label [2] | **98.27%** |
| *Our ConvBERT-base* | 98.14% |
| 9-fold CV, excluding *x* and *e* errors | |
| DMS, ELECTRA-base [3] | 97.84% |
| *Our ConvBERT-base* | **98.00%** |

Table 3: Accuracy results for different POS transformer models pre-trained on IGC and the accuracy of our transformer model when fine-tuned and evaluated in a comparable manner. [1] were reported in Daðason and Loftsson (2022), [2] in Snæbjarnarson et al. (2022), and [3] in Jónsson and Loftsson (2022).

language model during pre-training.

Figure 2 shows the accuracy improvements of the models for the more frequent lexical categories.

### 4.2 Transformer models and SOTA

In Table 3, we show previously reported results for transformer models pre-trained on the IGC, and the results of our transformer, a ConvBERT-base model trained and fine-tuned slightly differently compared to Daðason and Loftsson (2022) (see Section 3), evaluated in the same manner for comparison. Two of the papers cited in the table report results excluding the *x* and *e* tags, either both during training and evaluation or only during evaluation. These tags are used for unanalysed tokens and foreign words, respectively, and have the lowest category accuracies, the reasons for which will become apparent in Section 5. Not counting tagging errors for these two tags increases reported accuracy by 0.21 pp for our model. Excluding those tags from training, by fixing their weights to zero, increases the reported accuracy by a further 0.14 pp, because, in this case, the model is no longer able to assign these two tags erroneously to tokens.

The current SOTA is a *multi-label* model based on IceBERT-large[8] (Snæbjarnarson et al., 2022). Multi-label classification means that the tags are split into individual features, e.g., *lexical category*,

*tense*, *gender*, *number*, and the model is trained to predict each separately. Treating composite tags as multiple labels has been shown to improve POS tagging accuracy, especially when training data is scarce (Tkachenko and Sirts, 2018). Combining the predictions back into tags is dependent on knowledge about the composition of the tags. The results presented in Table 3 show that our ConvBERT-base model obtains SOTA results for single-label models applied to Icelandic.

## 5 Error analysis

In this section, we, first, present an analysis of the most frequent errors, and, second, the results of our analysis of the different sources of errors.

### 5.1 Most frequent errors

Table 4 shows the most frequent errors made by our transformer model. The list for the BiLSTM model is very similar, but with about double the accuracy degradation. The 12 most frequent errors are in fact six pairs of tags where the confusion between each pair occurs in either direction.

The most frequent confusion is $n$—$s$→$e$ (and $e$→$n$—$s$), or between foreign proper names and foreign words.[9] More than half, 0.04 pp for both error types, are due to words not seen during training. According to the MIM-GOLD tagging guidelines, compound foreign names should have the first word tagged as a foreign proper name ($n$—$s$), and then the rest of the name tagged as foreign words ($e$), except for names of persons and places that should have all parts tagged as foreign proper names ($n$—$s$). The tag $n$—$s$ is also used for abbreviations of foreign proper names, e.g., *BBC*. There are also some special cases that deviate from these rules (Barkarson et al., 2021). A significant portion of these tagging errors is indeed caused by annotation errors in the corpus (mostly $n$—$s$→$e$), as well as the fact that the application of the rules requires world knowledge that the models of course lack.

Confusion between adverbs and prepositions (which are annotated in MIM-GOLD as adverbs that govern case), i.e., $aa$→$af$ (and $af$→$aa$) are the next most frequent errors. Some of these tagging errors are due to cases where there is a clause between the preposition and the object, or where the

---

[8]IceBERT is based on a RoBERTa model (Liu et al., 2019).

[9]We denote a tagging error with $a$→$b$ where $a$ is the predicted tag and $b$ is the gold tag. The tag $n$—$s$ stands for a proper noun without markings for gender, number, or case.

| No. | Predicted tag → gold tag | Degradation in pp |
|---|---|---|
| 1. | *n—s → e* | 0.07 |
| 2. | *e → n—s* | 0.07 |
| 3. | *af → aa* | 0.05 |
| 4. | *aa → af* | 0.05 |
| 5. | *nheo → nhfo* | 0.03 |
| 6. | *fpheþ → faheþ* | 0.03 |
| 7. | *nveþ → nveo* | 0.03 |
| 8. | *nhfo → nheo* | 0.02 |
| 9. | *nveo → nveþ* | 0.02 |
| 10. | *ct → c* | 0.02 |
| 11. | *c → ct* | 0.02 |
| 12. | *faheþ → fpheþ* | 0.02 |

Table 4: The 12 most frequent tagging errors our transformer model makes. The rightmost column shows accuracy degradation in percentage points for each error type.

object has been moved to the front of the sentence. There also seem to be a fair number of annotation errors associated with this confusion between adverbs and prepositions.

A confusion between personal and demonstrative pronouns, *fpheþ→faheþ* (and *faheþ→fpheþ*), is caused by the antecedent being out of context or being a whole clause. Understanding the clause is often necessary to make the distinction. These are all the same word form, *því* ('it' or 'this, that'). For *því/fpheþ→faheþ*, we see some improvement in accuracy with the transformer model over the other models, but for *því/faheþ→fpheþ*, we notice the only case of lower accuracy for the transformer model compared to the others. The tags here are for neuter (*h*) singular (*e*) in the dative case (*þ*). There are identical confusions for the accusative and genitive cases, but those tokens are not as frequent.

The *c→ct* (and *ct→c*) errors are comparative conjunctions being marked as relativizers (a subordinating conjunction indicating a relative clause) and vice versa. Except for a few antiquated uses of *er*, these cases are all the word form *sem* ('as' or 'who, whom, that, which'). The conjunction *sem* subsumed *er*'s role as a relativizer in Old Icelandic. This language change was feasible due to their syntactic structures being identical (Kemmer, 1984). Semantically their function is similar, as one complements and the other modifies a noun phrase with the following clause. The

difference is this role of the relation. Therefore, the remaining tagging errors for *sem* are caused by a lack of syntactic and contextual information to make the correct prediction. Indeed, Loftsson et al. (2009) suggested that two tag categories be merged.

The errors *nheo→nhfo* (and *nhfo→nheo*), are confusions between the singular (*e*) and plural (*f*) forms of neuter nouns (*nh...*). When this error occurs, the context is usually not enough to determine the correct number. A wider context, previous sentences, or general knowledge is needed, and might even not be enough. Finally, *nveþ→nveo* (and *nveo→nveþ*) are confusions between the dative (*þ*) and accusative (*o*) cases of feminine nouns (*nv...*). The word that governs the case needs to be in the context, if it is omitted the distinction cannot be made. Moreover, if it can govern both cases, the required semantic information is unavailable.

One other group of errors should be mentioned, *∗→x*, where *∗* is any tag and the *x* tag denotes *unanalysed* tokens. This error is obscured because the predictions are distributed over many tags. These are tokens that contain spelling mistakes or constitute grammar errors and are the majority of the 2,777 tokens in the *unanalysed* tag category. Of the four models, the transformer does best with this tag category but is only predicting 58% correctly. Without changing how the spelling mistakes are annotated in MIM-GOLD or simply excluding sentences containing them, this will continue to be a source of about 0.12 pp accuracy degradation. As the corpus also contains tokens with such mistakes that are not annotated as *unanalysed* it would be in line with current practice to look to the intended meaning of these tokens and tag them accordingly.

## 5.2 Sources of errors

Manning (2011) discusses the generally perceived 97% token accuracy upper limit for POS tagging. At that time, those accuracy numbers had been reached for English, but Icelandic, a morphologically richer language with a very fine-grained tagset, had a long way to go. Rögnvaldsson et al. (2002) had earlier suggested 98% as the highest possibly achievable goal for Icelandic, because of inter-annotator disagreement. Manning reasons that the disagreement might actually be higher but says it is mitigated with annotator guidelines and

adjusting tag categories. Besides disagreement, subjectivity in annotation and the possibility of more than one right choice make up what Plank (2022) calls human label variation.

Manning samples errors the Stanford POS Tagger (Toutanova et al., 2003) makes when applied to a portion of the Penn Treebank corpus. He analyses the errors to try to understand if and how tagging accuracy could be further improved. He finds that the largest opportunity for gains is in improving the linguistic resources used to train the tagger. Before the initial release of MIM-GOLD, Steingrímsson et al. (2015) carried out an identical analysis on errors in both the IFD and MIM-GOLD when tagged with IceStagger. Their findings concurred with Manning's. We performed a similar analysis, though with a less detailed classification of the errors.

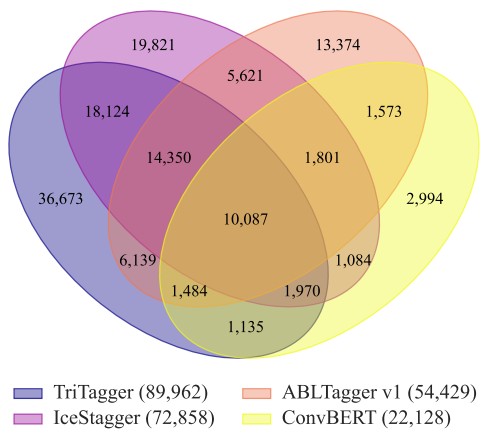

Figure 3: Venn diagram showing how prediction errors are shared between the four models.

Of the 1,000,218 tokens in MIM-GOLD, our transformer model makes 22,128 tagging errors. For 10,087 of these tokens, the three other taggers also make errors (see Figure 3), and for 5,526 of them, all four taggers agree on the predicted tag. From this set of errors, we drew a random sample of 500 for analysis. In this sample, we discovered 166 annotation errors, i.e., incorrect gold tags. For 150 of them, the taggers predicted the correct tag. Extrapolating to the superset gives us 1,658 tagging errors caused by gold errors ($\approx$0.16 pp). We also found 87 cases where the prediction error was obviously caused by there being insufficient context information ($\approx$0.09 pp), and 18 cases where it was likely caused by a spelling or grammar mistake ($\approx$0.02 pp). The last error class (spelling or grammar mistakes) is aggravated by the use of the

*unanalysed* tag (*x*) for such mistakes in the corpus. Table 5 shows the accuracy degradation for each of these error classes. Though we cannot draw conclusions from these findings about the frequency of these errors in the whole set of 22,128 errors, it is safe to assume these are the lower bounds of these error categories.

| Error class | pp |
|---|---|
| Annotation errors | 0.16 |
| Insufficient context | 0.09 |
| Spelling or grammar mistakes | 0.02 |
| Unexplained | 0.25 |
| Total | 0.52 |

Table 5: Estimated accuracy degradation in percentage points caused by each class in the set of prediction errors that all four taggers agree on.

# 6 Conclusions and Future Work

For Icelandic POS tagging, we have reached a point where individual error categories no longer stand out and annotation errors in the corpus are more pronounced, as well as inconsistencies stemming from human label variation.

Clear annotation errors can be corrected in the corpus, and the tagging guidelines and tag categories can be refined to remove some of the inconsistencies. Further gains can as well be squeezed out of the transformer model by using a larger model, i.e., ConvBERT-large instead of ConvBERT-base, increasing the vocabulary size, training it on the 2022 version of IGC that adds 549 million tokens, and fine-tuning the hyperparameters for the tagging model. Yet, on top of the annotator disagreement, there will always be errors because of a lack of information in the context, as well as the scarcity of examples to learn from for the long tail of infrequent tags.

For MIM-GOLD, that unsolvable part of the tagging errors seems to amount to less than 2 pp. Therefore, with a little more work, we should be able to confidently pass that 98% accuracy goal (when training and evaluating using the whole tagset) envisioned twenty years ago. A good starting point would be to search for and fix those estimated 1,658 annotation errors in MIM-GOLD, which are a subset of the tagging errors that all four models agree on.

To conclude, POS tagging for Icelandic is very close to being solved!

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
