# OpenReview forum: "Is Part-of-Speech Tagging a Solved Problem for Icelandic?"
_NoDaLiDa/2023/Conference — NoDaLiDa 2023_

### Official Review · Reviewer_SuLU · 2023-02-27
**A small study for a small audience**

**Rating:** 5
**Confidence:** 3

**Review:**

This paper investigates the efficacy of four POS-tagging models for Icelandic, and discusses what remains before the problem can be considered solved. It finds that the most current model is indeed the best, and that there is still a little bit of room for improvement.

The experiments seem overall thoroughly done; the authors have not only tested the models but also investigated the errors in detail, and drawn conclusions about what this might mean for future progress.

When discussing the somewhat narrow topic of POS-tagging, in the somewhat narrow language of Icelandic, there is of course a challenge in targeting one's audience. Should the paper be written for those intimately familiar with Icelandic POS-tagging, or for a more general NLP audience? This paper contains quite a few terms and techniques that are hastily or unclearly explained. I consider myself at least somewhat familiar with POS-tagging, but I must admit that passages like "span-based dynamic convolution instead of self-attention heads" are not obvious to me, and probably also not to a general NLP audience. There are also several places where tags are used without explanation, which makes understanding more difficult for those not familiar with the specific tagset. Obviously not everything can be explained in detail, but perhaps in some cases it would be better to make either a more thorough explanation, or leave it out and rely on the references.

I have to conclude that while the work seems adequately done, the description could perhaps have been improved to be a little clearer outside a very small circle. I am also not fully convinced that this could not have been a short paper.

Minor comments:

"three previous, as well as
the current, state-of-the-art data-driven Part-of-Speech tagging model types"
How were they "previous"? It sounds odd to think of "state-of-the-art" as something completely black-and-white. Should this be taken to mean that they held some sort of record? Surely there's more than one Icelandic corpus, and therefore no one objective record holder?

"it is safe to assume that increased length results in increased complexity"
Should we take this to mean that per-token accuracy decreases with sentence length?

**Paper Type:**

Long paper

---

### Official Review · Reviewer_72BL · 2023-03-06
**The authors of this paper train and test four data-driven POS-tagging models for Icelandic and conclude that, with very little additional effort, POS tagging could be called a solved problem for Icelandic. However, the text types of the gold standard are not described and so it is impossible to say, is the POS tagging really solved for the majority of text types of Icelandic.**

**Rating:** 7
**Confidence:** 4

**Review:**

The authors of this paper have trained and tested four data-driven POS taggers (TriTagger, IceStagger, ABLTagger and ConvBert) on a new large gold-standard corpus (ca 1 million tokens). The best model, ConvBERT-base achieves 97.79% token accuracy and 73.43% sentence accuracy. Basing on error analysis, the authors infer that after fixing errors in the gold data and using a larger model, it should be possible to pass the 98% accuracy goal. Their conclusion is that POS tagging problem for Icelandic is very close to being solved.

However, "Icelandic" is, in this article, limited to the gold standard corpus. The corpus is quite big, containing ca 1 million tokens, but the text classes (13 altogether) of this corpus are not described or even listed in the article and so it is difficult to say, what language varieties does it contain and how representative it really is, does the gold data contain also spoken language etc.

**Paper Type:**

Long paper

---

### Official Review · Reviewer_yBt2 · 2023-03-10
**POS tagging for Icelandic is almost solved, this paper sheds a light on remaining problems**

**Rating:** 7
**Confidence:** 4

**Review:**

This paper investigates whether POS tagging for Icelandic is a solved task. They do this by evaluating a variety of taggers on the latest version of MIM-GOLD. They propose a new tagger based on convbert with some tiny improvements. They conclude that there is still a small margin for performance improvement, but that the task is very close to being solved. The remaining errors mostly stem from the data instead of the taggers.

Strengths:
- Comprehensive comparisons to other taggers
- Qualitative analysis seems comprehensive
- It is important to stop trying to improve scores and take another look at our tasks/data from time to time.

Weaknesses:
- Not sure whether to interpret this as a short or long paper. With the current content, it makes for a fine short paper, but the contributions are a bit limited the small side for a long paper.
- Other datasets are available but are ignored, (e.g. UD, IFD)
- The wrong style-files are used, I cant refer to line numbers

Questions:
- you do not touch upon ambiguity, is this not a problem for POS tagging of icelandic?
- Do you see it as a future improvement to reannotate the Unanalyzed class? Or is this in contrast to the whole annotation scheme?( I guess literal meaning are prioritized as opposed to intended meanings)
- Does increased length actually lead to increased complexity (i.e. lower performance)?
- The number errors can perhaps be resolved by splitting the labels, which you seem to suggest is not beneficial at the end of 4.2 (but note that there are many confounding factors here!, splitting sounds like a better fit to the task to be honest)
- How do you calculate accuracy per tag in Figure 2?, is this just a binary task then?

ps. in the last sentence: closed->close

**Paper Type:**

Long paper

---

### Decision · Program_Chairs · 2023-03-17

Accept